# Adolescent Body Dissatisfaction in Contrasting Socioeconomic Milieus, Coming from a French and Luxembourgish Context

**DOI:** 10.3390/ijerph17010061

**Published:** 2019-12-20

**Authors:** Faustine Regnier, Etienne Le Bihan, Christine Tichit, Michèle Baumann

**Affiliations:** 1Alimentation et Sciences Sociales, Université Paris-Saclay, INRA, UR, ALISS, 94205 Ivry-sur-Seine, France; faustine.regnier@inra.fr; 2Institute for Research on Sociology and Economic health Inequalities, University of Luxembourg, L-4365 Esch-sur-Alzette, Luxembourg; etienne.lebihan@uni.lu; 3CMH—INRA, CNRS ENS-EHESS, 75008 Paris, France; christine.tichit@inra.fr

**Keywords:** adolescence, BMI, body dissatisfaction, children’s body image scale, social differences

## Abstract

*Purpose*: To analyze the relationships between body mass index (BMI), ideal body, current declared body shape, and gap between ideal and declared body shape, and the associations that these have with social and cultural factors among 329 adolescents (11 to 15 years i.e., at two stages of adolescence, the early and late adolescence), attending an international school in Luxembourg, and 281 from Paris. *Methods:* A cross-sectional survey was conducted using an online questionnaire. Missing data were addressed using the data augmentation method in a Bayesian framework. *Results*: For both sets, higher the BMI and bigger their current body shape (CBS), the slimmer their ideal body shape, especially for those who perceive a link between body shape and beauty. For girls, slimness is a shared ideal; for boys, older they are, more they want a muscular body shape. Most students want slimmer bodies, but in affluent or intermediate social milieu students in relations to identification to personalities such as celebrities, while students from modest milieus, this is expressed in relation to success in love. In addition, they declared that their “talk diet with friends” were associated with large gap between ideal and declared body shape. *Conclusions:* A social control norm was revealed involving a displacement of values affecting body weight and health in the late stage of adolescence to early adolescence, especially for boys.

## 1. Background

During adolescence, psychological factors and socioeconomic status can have an especially large impact on health issues such as overweight and obesity [1,2]. These can be responsible for numerous health complications later in life, such as diabetes, cancer, arthritis, other chronic diseases, and increased risk of mortality [3]. Body image is a cultural and psychological construct whose importance increases as young adolescents become more body conscious, primarily because of physical changes associated with puberty. Poor body image among children and adolescents can have severe health implications, causing reduced levels of physical activity [4], unhealthy eating behaviors [5], and mental health problems such as depression [6].

Teenagers are an at-risk population regularly exposed to propaganda and conventional norms concerning beauty and leanness in the media, at school, and in their peer groups [7,8]. Body dissatisfaction is a significant concern in itself, as it is related to a wide range of negative outcomes including depression and eating disorders [9]. The discrepancy between current body shape and ideal body may be a source of dissatisfaction, engendering the desire for change. Reciprocal relations between body satisfaction and self-esteem are stronger in late adolescence and emerging adulthood than at earlier and later stages [8].

A negative perception of one’s own body can trigger dietary behavioral problems [10,11], contributing thereby to overweight and poor health. However, there is insufficient evidence concerning the influence of social and cultural determinants on body satisfaction [11]. Moreover, the prevalence of negative body image increases in early and mid-adolescence. Understanding the factors that affect how a teenager perceives his/her “current body shape” (CBS) and conceives of an image of the “ideal body” s/he wishes to have (IB), as well as the relationship these body images have with body norms, should be research priorities. Finally, gender differences appear to be a major dimension [12]. To measure the discrepancy between “ideal body” and the representation ′current body shape′ which may be source of dissatisfaction and the expression of expected change, we opted for the instrument of the children’s body image scale widely used, and scientifically validated [13].

The foundations of positive body image patterns are developed during young adolescence [14,15,16]. Protective factors in these regards include healthy eating patterns and regular physical activity [17], acceptance by peers and family, and better social relationships [18]. Being overweight or obese in youth can lead to orthopedic disorders and psychosocial difficulties, such as poor self-esteem or depression [19]. To combat overweight and obesity, governments have launched nutritional plans under which public health campaigns are held and nutritional recommendations disseminated among the populace, particularly in schools [20]. Examples include the national action plan “Gesond iessen—méi bewegen” (Eat healthy, move more) in Luxembourg and the National Nutrition and Health Program in France.

In order to grasp this evolution through the two stages of adolescent development, examining perceptions of body image among young adolescents from two different milieus (intermediate and affluent vs. modest), from a French and Luxembourgish context, the aims of our study are to: (1) analyze the relationships among body mass index (BMI), CBS, IB and the gap between ideal and declared current body shape (GIDB) and their associations with social and cultural factors; and (2) identify gaps or flaws in data related to BMI, CBS, and IB in order to identify the advantages and limits of BMI and the children’s body image scale as measurement instruments.

## 2. Methods

### 2.1. Samples

Among 329 adolescents attending school in Luxembourg and 281 in Paris participated. They were aged 11 to 15 years, or from the beginning of secondary school (the second stage of the educational cycle in European countries) through to year four of secondary school (followed by entry into the third educational cycle).

We administered a questionnaire in these two schools to compare and contrast body image according to their different social and cultural contexts: in Paris, the students mainly came from low-income families and their parents were of African origin. In this Parisian district, in 2015, the median income was 19.14 euros per consumption unit (CU) vs. 26,431 for Paris as a whole, while in 2014, the unemployment rate among 15–64-year old people was 16.6% vs. 12% for Paris and the poverty rate was 24.6% vs. 16.2% for Paris [21].

In the second school, an international school in Luxembourg, students mainly came from intermediate to affluent milieus, and from eight different European countries (mainly France, UK, Denmark, Germany, and Italy). The socioeconomic environment in Luxembourg (in general) is drastically different from that of the Parisian district considered: in 2014, the median income was 34.3 euros per CU (Consumption Unit) the unemployment rate was 6%, and the poverty rate 19% [22].

### 2.2. Ethical Aspects and Administrative Arrangements

The investigation conforms to the principles outlined in the declaration of Helsinki. The study protocol was approved by the ethic committee of the Institute for Research on Sociology Economic Inequalities (R-STR-3064-02-Z). Medical and psychological professionals were involved in the preparation of the questions, ensuring their face and content validity. The school boards of directors gave their approval, and parents signed consent forms after a meeting explaining the purpose of the study and the data collection procedure. The students’ participation was voluntary, and students were explained the survey (aims, data collection and anonymity process).

### 2.3. Design and Translation of the Questionnaire

Cross-sectional data collection was conducted in May 2016 with students who completed an online questionnaire available in French, German, and English. We created the items of our instrument after a qualitative pilot study among adolescents’ focus groups. Although Luxembourg is multilingual and very culturally diverse (more than 170 different nationalities), the official languages are Luxembourgish, but also French and German. English, then, is the most common foreign language taught in the international school where we organized this survey. Each version had been translated, back-translated, and then proofread by native-speaking professional translators. In order to avoid mis-comprehension of the questionnaire, because of the differences of maturity between children of 11 and of 15 years old, we have tested the self-administrated questionnaire in focus-groups before its administration, and modified it according to the students comprehension and reactions.

### 2.4. Measures

*Gap between ideal and declared body shape (dependent variable).* In order to determine the gap between his/her desired and current perceived body, each young adolescent was asked to choose the image s/he felt best illustrated her/his current body shape (CBS) and the image that best corresponded to his/her ideal body (IB). The children’s body image scale (CBIS; Appendix A) was applied; this is a widely used and scientifically validated instrument [13]. This pictorial scale contains seven pictures each of boys’ and girls’ bodies, representing standard percentile curves for BMI, that is, differences in adiposity between girls and boys ranging from A to G in ascending order of weight. We attributed values between one and seven to each of the responses for IB and CBS (A = 1, B = 2… etc.,), and then computed the difference between them. This means that the difference could theoretically vary between −6 and +6, with a negative difference indicating that the child wishes to have a lower weight-to-height ratio than that which he/she reported.

*Demographic and social characteristics.* Characteristics gathered were gender and age of the participants, educational level (first to fourth year of secondary school), mothers’ and fathers’ occupational category, parents’ birth countries, attitudes toward media discourse, relationships with friends, level of physical activity, and self-representation of body form.

### 2.5. Statistical Analyses

*Relation between CBS and BMI.* First, we sought to associate answers on the children’s body image scale with the students’ BMI. To achieve this, we adjusted a BMI regression model on CBS in interaction with gender and used this model to predict the values sought by girls and boys. Two analyses were conducted, respectively based on the following assumptions: (1) CBS is a continuous variable, and the relationship between CBS and BMI is linear; and (2) CBS is a qualitative variable, and its relationship with BMI has no imposed form.

*Determinants of GIDB.* We established a GIDB adjustment model taking age, gender, and BMI into account using multiple linear regression; this was in order to avoid attributing direct effects of a variable to those of other variables with which it may be linked. The reason for considering BMI is that it was deemed likely that an adolescent with a high weight/height ratio would be more likely to desire a decrease in this ratio, and so their GIDB would be negative. On the other hand, we expected that adolescents with low weight/height ratios would want a higher ratio for their ideal body than their CBS, so their GIDB would be positive.

We then evaluated the links between the GIDB and each of the other variables, considering the possible impact the respective social context could have on these links by introducing an interaction effect between each explanatory variable and the students’ socio-economic characteristics.

*Taking account of missing values.* Participants did not answer questions relating to their height and weight, which resulted in a considerable amount of missing data on these variables. Consequently, we chose to adjust the above model using a data augmentation procedure implemented in a Bayesian framework [23]. This approach has two major advantages: (1) It allows all available information to be taken into account, and (2) it provides unbiased estimates of the parameters, providing the missing values are MAR (missing at random) [24]. All analyses were performed using R and OpenBUGS software (MRC Biostatistics Unit, Cambridge, United Kingdom) [25,26] (with package R2OpenBUGS [27]).

## 3. Results

Table 1 reveals the socioeconomic background of the students. In the international school in Luxembourg, over 70% of the parents were born in European countries. In the French school, over 50% of the parents were born in Africa: 30.5% of mothers and 29.4% of fathers in sub-Saharan Africa and 20.7% of mothers and 23.7% of fathers in the Maghreb. Another indicator of the social background of students is the parent’s employment status: the most frequent professional status of the fathers in the international school was executive (47.8%), whereas it was manual worker in the low-income school (34.0%).

As seen in Table 2, in both schools, GIDB (= IB − CBS) scores were approximately equal, which is explained by the fact that students from low-income milieus also showed higher average IB. Students’ average size was similar in both schools, but weight averages and BMI were slightly higher among students from modest milieus, consistent with their higher CBS average. There were considerable missing data (see column “#NA”) concerning the size and weight variables in particular, resulting in 57.7% of BMI missing values. Meanwhile, the percentage of CBS data that is missing is low (10.4%) compared to that for IB (26.9%).

As seen in Table 3, for students from modest milieus, missing data for BMI were associated with age (the younger the adolescent, the greater the likelihood that BMI was missing). An association with social context was also found (the proportion of missing data was higher among students from intermediate or affluent milieus). For CBS, the younger the adolescent, the higher the probability of missing data, in both milieus. More boys than girls did not provide IB data, and overall, the proportion of missing data was higher among students from intermediate to affluent milieus.

Figure 1 shows that for all respondents, BMI (Figure 1a) increases when moving from one CBS category (A–G in the children’s body image scale [13]) to the next; the slope is generally the same for girls and boys (approximately 0.5 BMI points). Overall, a positive relationship exists between BMI and CBS (Figure 1b); however, locally, we observed a slight decrease in predicted BMI between categories C, D, and E. This observation was not expected, and seems counter-intuitive. A linear relationship exists; it seems that young adolescents with BMIs slightly higher than the average tended to choose category C, which depicted a somewhat thin physique, and that, inversely, those with a somewhat lower BMI chose category E, showing a somewhat bigger body. Again, the predicted values for boys and girls are similar, and the decline in predicted BMI values between C, D, and E is observed in both genders.

See the adjustment model (Appendix B).

As seen in Table 4, regardless of the school, young adolescents who stated that body shape is important to be considered attractive wished to be slimmer than they reported being. However, opinions concerning the usefulness of sports (on weight, beauty, well-being, pleasure) had no significant connection with GIDB.

Students from intermediate to affluent milieus who said they wanted to change their height had an ideal body image that was lower than their reported body shape, and also wished to have a slimmer body. Further, those who wished to resemble media personalities desired an ideally toned physique that was lower than their reported shape. Meanwhile, students from modest backgrounds who wished to improve their body shape also generally wished to change their weight; for these students, the perception that beauty facilitates success in love is related to the wish to have a slimmer body. The attitude of students who talked with friends about diet is associated with negative GIDB, in comparison to those who never discuss body image-related topics.

## 4. Discussion

The higher their BMI and CBS, the more students’ IB is slimmer than their current shape. Younger adolescents have been more susceptible to nutritional education than older students; such a disposition may mean that information broadcast by media (TV, Internet) regarding the ideal of slimness can affect this group to a greater degree, or may conversely be a positive outcome of Luxembourgish and French public health policies among the young generation; evaluating the respective role of each sphere (national public health programs vs. media influence) is difficult.

Our results show that from early adolescence, there is a strong integration of body norms associated with slimness; this corroborates the results of similar work performed in the Anglo-Saxon context [28,29]. This observation highlights two discourses that cover the representations adolescents have of their bodies: the influence of the media, which promotes the sociocultural and sexual ideal of a thin body; and the awareness of preventive messages related to fighting “juvenile obesity.” These attitudes toward health are present among our study participants’ adolescents, independent of social or cultural background, in a context of similar nutritional policies, in particular school-level health-promotion programs such as the French National Plan for Nutrition and Health (PNNS) that advocate the need for regular physical activity and a balanced diet rich in fruit and vegetables. Indeed, the combination of several levels of intervention concerning diet and physical activity makes overall efforts more effective [30]. In the same line, it is surprising that opinions concerning the usefulness of sports turned out not to be related to body image. In this matter, gender differences are more significant than social ones; when it comes to body work, in all milieus, boys quickly establish a link with the usefulness for improving one’s CBS of physical exercise, whereas girls are more centered on body work through diet.

Other interesting findings concern the effects of sociocultural and gendered inequalities, such as the impact of peers on GIDB. In both schools, students who stated that body shape is important in order to be beautiful (or handsome) desired a thinner body shape. Similarly, among boys, the older they were, the more muscular they wished to be. During early adolescence, girls and boys become more keenly aware of their gender, and they adjust their behavior or appearance to conform to perceived norms [31]. Other effects, such as that of media, may also intervene. During the early stage of adolescence, parents are the dominant source of information; however, peers and the media become preponderant at the late stage, in which appearance is a major concern in relation to peer relationships. This attitude was observed among affluent and intermediate-milieu students; those who wished to resemble media personalities wanted to be slimmer. At this stage, the impact of gendered beauty stereotypes (lean and defined muscular body for men) is already present in their discourse, and this means that the relevance/importance of the role of the media is greatest within the late stage of adolescence (over 15 years old) [28]. Certain values affecting the body and health during late adolescence were displaced during the early adolescence stage, and corroborate previous results on the associations between BMI, body dissatisfaction, and shape concern in relation to gender in late adolescence [32].

Adherence to beauty stereotypes creates recurrent problems that interfere with identity building and psychologically disrupt adolescents as they age, particularly those prone to eating disorders, such as anorexia, bulimia, or leading to overweight or obesity [30], which can occur very early in life. In late adolescence, girls tend to be at greater risk of negative health outcomes related to weight than boys, including depression; this vulnerability derives in part from profound anxieties concerning body image that are fueled by social and media stereotypes of beauty [33]. Among students from modest milieus, discussing food with friends was linked to a desire to have a slimmer body shape, suggesting the development of an interest in body work through diet. Believing that beauty facilitates success in love was also found to be related to wanting a slimmer body. Regarding social and romantic relationships, physical appearance is important for young adolescents, who are at risk for adverse psychological and social experiences such as discrimination [34] and bullying [35] and who are becoming more aware of gender and sexual relations.

Current body shape was slightly larger among students from low-income milieus, but GIDB (= IB − CBS) was equal between the samples, which is explained by the existence of higher IB among students from low-income families. A link between body shape and parents’ social situation has been highlighted in previous work on the unequal social distribution of overweight children [30,33]. We suggest that our result is related to the impact of differences in social status between parents. Cultural belonging could also be linked; over 50% of parents from the French school were born in Africa, and certain African cultures traditionally value more adipose bodies especially among women [36]. A recent study showed that the home food environment explains a high portion of the link between low socioeconomic status and lower consumption of healthy foods; both social (mealtime structure) and physical aspects (food availability) of the home food environment are strongly associated with the consumption of healthy or unhealthy foods [37].

### 4.1. Limitations and Strengths

The analyses of the missing values and the profiles of the non-respondents revealed some limitations of the current approach in capturing declared BMI, current self-perceived body shape, and ideal body, and confirmed the necessity of using an adjusted statistical analysis model. Although using the Internet to apply such measures is recognized as appropriate for young adolescents. Indeed, the web-format is considered equivalent to the paper version [38]. Unfortunately at this age, some of them did not to want to answer the questions asked by the adults. Also the study of the missing data can be revealing hidden hypothesis. The missing data (57.7% on BMI) shows that the students generally do not know their weight and height, and that they do not recognize themselves in the pictures (26.9% IB, 10.4% CBS) presented, and this could be linked to the fact that they are pubescent. These observations suggest that it is necessary to remain cautious about the results of research performed using the children’s body image scale; although it is widely used to measure body satisfaction in children, there are some limitations to the international scientific literature in which it has been applied [13]. Boys more than girls seem to consider their body perception an intimate domain and studies have shown that they rarely elaborate about weight [39]. Girls may be more able to discuss it frankly, since female bodies are much more objectified in the media.

Alongside the above limitations, however, in addition to the results already discussed our approach confirms that the method used to address the missing data (i.e., data augmentation; see “Statistical Analyses”) was valid; access to such methods is important, as the presence or absence of data is associated with the values of the observed data, and the usual method of discarding observations with missing data would have led to biased results.

A positive relationship between BMI and CBS exists; students who consider themselves too skinny choose images of bodies that are slightly stronger, while those who consider themselves too big choose slightly leaner images. Our findings corroborate previous observations on the difficulty of estimating self-declared weight and size [40] and highlight significant body-image dissatisfaction, especially among girls [41]. The use of body shape perception remains a good method of correcting the distortions of the measure of BMI perceptions in young adolescents, supporting its use in future studies [42].

### 4.2. Practical Implications

Some points raised by the young adolescents appeared earlier than expected in relation to the United Nations Children’s Fund (UNICEF) (2011) definitions of the periods of early adolescence (10–14 years) and late adolescence (15–19 years) [31]. Our study confirms the differences between these two stages, but suggests that the content of interventions designed for each stage needs to be adapted to the particular sociocultural values that young European teenagers face and hold.

From this perspective, the children’s body image scale (CBIS) presented limits to visualizing one’s present body, in particular for younger adolescents, and also one’s ideal body, especially for boys. Focusing on body weight hampered our ability to grasp and analyze boys’ specific concerns, such as muscles. The consequent hypothesis is that this scale does not allow us to grasp the diversity of concerns and views that young adolescents have in regard to their bodies (not only weight but also size). The presence of missing values for IB in boys confirms that they have more difficulty than girls recognizing themselves in the images proposed by the CBIS: the reduction of body shape to distribution of adiposity does not reveal the bodily criteria important to them. Based on our findings, we propose that other measures for boys be used to help grasp teenagers’ gender-specific concerns about body shape, for example, “assessing muscularity among other physical features” [12]. This point is less documented in the literature regarding body dissatisfaction, and is often considered through the lens of adiposity. Separately, and broadly, targeted interventions to increase physical activity seem relevant at a stage when sedentariness is increasing [43].

## 5. Conclusions

The relationships between IB, CBS, BMI, and social and cultural values, as reflections of cognitive and social development along with physical and psychological upheavals, revealed attitudes common to the same generation but differing considerably between genders, and stages of adolescence. But ways of socialization among students tend to minimize cultural differences.

A social control norm for body weight in adolescents was identified, with a displacement of values affecting the body shape and health from the late to the early adolescence stage. To better formulate and implement preventive interventions and nutritional recommendations, new instruments better adapted to psychological and physical changes in younger generations must be developed.

## Figures and Tables

**Figure 1 ijerph-17-00061-f001:**
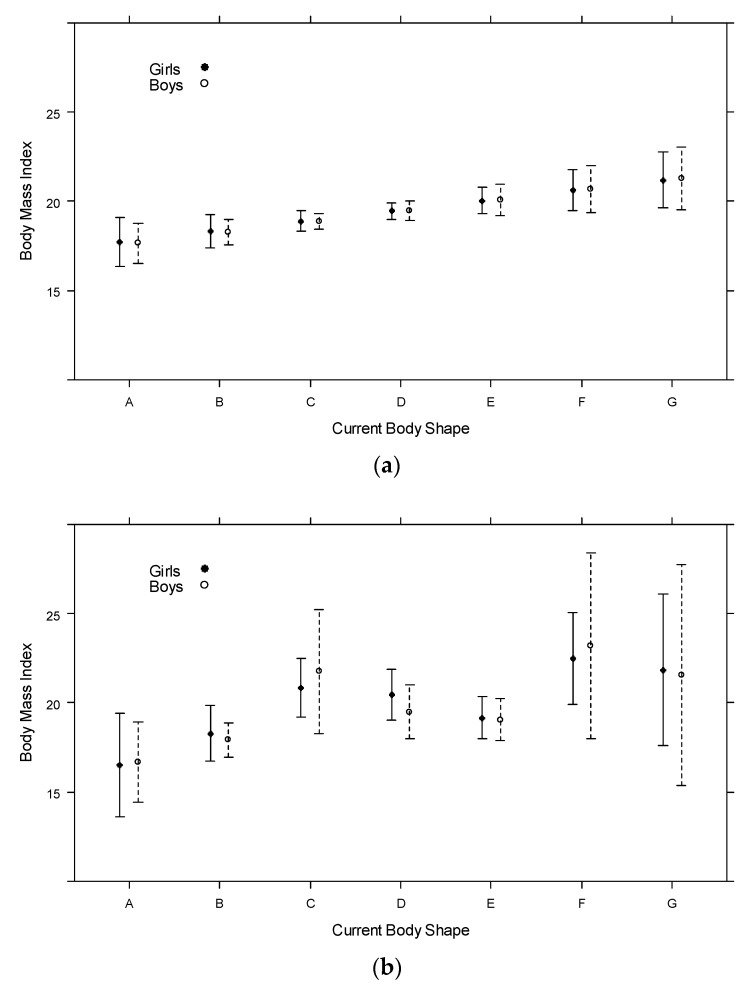
Predicted BMI values (and their credibility intervals) as a function of the current body shape (CBS) class, by gender of student, (**a**) assuming a linear shape relation, (**b**) not imposing any specific form to the relationship.

**Table 1 ijerph-17-00061-t001:** Socio-demographic characteristics of the participants.

Socio-Demographic Characteristic		Luxembourgish School *n* = 329	French School *n* = 281
		% or mean (SD)	% or mean (SD)
Age		13.7 (1.7)	13.7 (1.3)
Gender	Male	46.8	51.1
	Female	53.2	48.9
Mother’s country of birth	France	17.7	20.0
	Other European country	70.3	2.9
	African country	2.5	51.3
	Other regions of the world	9.5	25.8
Mother’s professional status	Executive	23.0	6.7
	White collar	50.0	45.5
	Blue collar	0.4	17.9
	Non-working	26.7	29.9
Father’s country of birth	France	15.3	17.9
	Other European country	71.0	3.0
	African continent	5.1	52.9
	Other regions of the world	8.6	26.2
Father’s professional status	Executive	47.8	16.6
	White collar	40.9	24.7
	Blue collar	3.6	34.0
	Non-working	7.7	24.7

**Table 2 ijerph-17-00061-t002:** Descriptive statistics of body variables. Mean (SD).

Body Variable		Luxembourgish School		French School	
	# NA ^1^	*N*	Mean (SD)	*n*	Mean (SD)
Height (m)	273	212	1.61 (0.11)	139	1.61 (0.11)
Weight (kg)	297	186	48.4 (12.2)	141	50.2 (11.7)
BMI (kg m^−2^)	360	160	18.8 (2.9)	104	19.5 (3.3)
IB (1;7)	168	234	3.34 (1.6)	222	3.53 (1.5)
CBS (1;7)	65	300	3.49 (1.5)	259	3.77 (1.5)
GIDB (−6;6)	168	234	−0.27 (1.7)	222	−0.27 (1.7)

^1^ Number of missing values.

**Table 3 ijerph-17-00061-t003:** Relationships between missing values for body mass index (BMI), current body shape (CBS), and ideal body (IB) variables, and gender, age, and school; % or mean (SD).

Variable			Luxembourgish School			French School			All		
			Missing	Not Missing	*p* ^1^	Missing	Not Missing	*p*	Missing	Not Missing	*p*
BMI											
	Gender	M	49.7	50.3	0.597	61.3	38.7	0.690	54.8	45.2	0.391
		F	53.3	46.8		64.3	35.7		58.6	41.4	
	Age		13.5 (1.7)	13.9 (1.6)	0.024 *	13.7 (1.3)	13.7 (1.3)	0.891	13.5 (1.5)	13.8 (1.5)	0.104
	School	Lux							51.4	48.6	0.005 **
		French							63.0	37.0	
CBS											
	Gender	M	8.6	91.4	1.000	10.2	89.8	0.143	9.3	90.7	0.394
		F	0.1	99.9		4.9	95.1		7.1	92.9	
	Age		13.7 (1.7)	13.7 (1.7)	0.835	13.0 (0.1)	13.7 (1.3)	0.002 **	13.4 (1.5)	13.7 (1.5)	0.163
	School	Lux							8.8	91.2	0.771
		French							7.8	92.2	
IB											
	Gender	M	37.7	62.3	0.000 ***	26.3	73.7	0.036 *	32.7	67.3	0.000 ***
		F	18.8	81.2		15.4	84.6		17.2	82.8	
	Age		13.5 (1.7)	13.7 (1.7)	0.273	13.5 (1.1)	13.7 (1.3)	0.269	13.4 (1.5)	13.7 (1.5)	0.136
	School	Lux							61.7	38.3	0.032 *
		French							44.4	55.6	

^1^*p*-Value of chi-squared test or student’s *t*-test. Significant *p*-Value: * *p* < 0.05; ** *p* < 0.01; *** *p* < 0.001.

**Table 4 ijerph-17-00061-t004:** Relationships between opinions on body matters and gap between ideal and declared body shape (GIDB).

Opinions on Body Matters		Luxembourgish School			French School		
		est. ^1^	PSD ^2^	*p* ^3^	est.	PSD	*p*
Sports improves health and helps people feel good	Yes	0.235	0.320	0.466	0.225	0.320	0.480
	No	0			0		
Sports helps make people handsome, muscular, and popular	Yes	0.012	0.248	0.964	−0.110	0.262	0.669
	No	0			0		
If you could change something about yourself, what would it be?							
Nothing	Yes	0.307	0.238	0.199	0.403	0.239	0.093
	No	0			0		
My weight	Yes	0.269	0.239	0.259	−0.470	0.235	0.044 *
	No	0			0		
My height	Yes	−0.656	0.229	0.004 **	0.037	0.243	0.879
	No	0			0		
In your opinion, what makes a person handsome?							
Body shape	Yes	−0.503	0.218	0.023 *	−0.610	0.228	0.008 **
	No	0			0		
Does beauty facilitate success in love?	Yes	−0.355	0.223	0.108	−0.522	0.226	0.022 *
	No	0			0		
Do you wish you resembled people in the media?	Strongly agree	−1.028	0.481	0.033 *	−0.734	0.436	0.092
	Agree	0.154	0.316	0.626	−0.304	0.343	0.385
	Disagree	−0.544	0.354	0.125	0.496	0.346	0.156
	Strongly disagree	0			0		
Do you often talk with your friends about food you like (or dislike) to eat?	All the time	−0.459	0.462	0.321	−0.492	0.403	0.224
	Often	−0.340	0.331	0.306	−0.570	0.346	0.101
	Sometimes	−0.474	0.288	0.103	−0.716	0.330	0.029 *
	Never	0			0		

^1^ Estimate. ^2^ Posterior standard deviation. ^3^ Significant *p*-value: * *p* < 0.05; ** *p* < 0.01.

## Data Availability

Data are available from the INRA and from the Institute for Research on Sociology and Economic Inequalities for researchers who meet the necessary criteria to access to the data. Researchers may contact the Institut National de la Recherche Agronomique (INRA) management board (see authors’ addresses).

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
