# Peer review of "Adolescent Body Dissatisfaction in Contrasting Socioeconomic Milieus, Coming from a French and Luxembourgish Context"

_ijerph, 2019, doi:10.3390/ijerph17010061_

Round 1

Reviewer 1 Report

Summary, Clarify the stages of adolescence, is put into practical implications

Too many keywords, review in guide for authors. Although it meets the criteria between 3-10, I would do a review.

Introduction

The use of questionnaires in the age range 11-15 can produce some errors, especially in the reading comprehension of the questions and in the maturity between the children of 11 and adolescents of 15. For this it would be interesting to justify that the scales used do not They have given problems of understanding.

Method

As for the use of the questionnaires in different languages, it would be interesting to put the validated versions in each language. They put translation information without citing official sources.

Results

ok

Discussion

ok

In the sections of practical implications and limitations it refers to issues discussed above such as early adolescence (10-14 years) and late adolescence (15-19 years). The study focuses on the first strip and has given some problems, lack of data. This would lead us to rethink the results obtained, although it has been justified very well.

Author Response

1st reviewer. Comments and Suggestions for Authors

Summary

Clarify the stages of adolescence, is put into practical implications

To analyze the relationships between BMI, ideal body, current declared body shape, and gap between ideal and declared body shape, and the associations that these have with social and cultural factors among 329 adolescents [11 to 15 years, i.e at two stages of adolescence (early / late), and attending an international school in Luxembourg, and 281 from Paris.

We thank you for this comment. We have specified in the summary and the introduction the two stages of adolescence we took in account.

Too many keywords, review in guide for authors.

Thank you for this observation. We have selected 5 key words. Key words : Adolescence ; BMI; body dissatisfaction; Children’s Body Image Scale; social differences

Introduction

The use of questionnaires in the age range 11-15 can produce some errors, especially in the reading comprehension of the questions and in the maturity between the children of 11 and adolescents of 15. For this, it would be interesting to justify that the scales used don’t have given problems of understanding.

Thank you for this comment. We have specified in our method: ‘In order to avoid mis-comprehension of the instrument, due to the differences of maturity between children of 11 and of 15 years-old, we have tested the questionnaire in focus-groups before its administration, and modified it according to the students’ comprehension and reactions’.

Method

As for the use of the questionnaires in different languages, it would be interesting to put the validated versions in each language. They put translation information without citing official sources.

Thank you for this remark. We don’t use validated versions of the literature. We created the items of our instrument after a qualitative pilot study among adolescents’ focus groups. Although Luxembourg is multilingual and very culturally diverse (more than 170 different nationalities), the official languages are Luxembourgish, but also French and German. English, then, is the most common foreign language taught in the International school where we organised this survey. We added this sentence in our method part.

Results. ok

Discussion. Ok

In the sections of practical implications and limitations.

it refers to issues discussed above such as early adolescence (10-14 years) and late adolescence (15-19 years). The study focuses on the first strip and has given some problems, lack of data. This would lead us to rethink the results obtained, although it has been justified very well.

 We thank you for this pertinent remark. Indeed, we think that taking into account missing data with the data augmentation method in a Bayesian context is particularly suitable here.

2th reviewer. Comments and Suggestions for Authors

The interesting manuscript based on a cross-sectional online questionnaire survey for adolescents [11 to 15 years], attending an international school in Luxembourg (329), and 281 from Paris. However, the details of the questionnaire did not be shown.

Thank you for your proposition. For us, the items inside the questions are in the tables. But if the editor agrees to add the French questionnaire, in the supplementary file, we will do it. Alternatively, questionnaires can be sent upon request to the authors. Concerning the validate Children’s Body Image Scale, you find it in a supplementary material (CBIS); we added an APPENDIX 1 with different pictures of girls and boys.

Finally, the authors concluded ‘A social control norm was revealed involving a displacement of values affecting body weight and health in the late stage of adolescence to early adolescence, especially for boys’ in the abstract.

Thank you

Comments:

The title contents “contrasting cultural” from “a French and Luxembourgish context”, however, I did not think the difference of cultural between French and Luxembourgish are the main point in this manuscript. The socioeconomic difference between these 2 groups of adolescents were indeed. Thus, the title and discussion need to re-consider or revise.

We thank you for this thought full observation. The title has been specified in favour of « in a contrasting socioeconomic milieus, from a French and Luxembourgish context ».

We have revised the words used in the discussions.

The authors deal with the missing data with the data augmentation method in a Bayesian framework. I think need a sensitivity analysis: comparison with and without the missing data augmentation method. I suggest to put the results in the supplementary file.

Thank you for your suggestion. We created at the end of the manuscript an APPENDIX 2 supplementary material with the description of the results and the tables.

Minors: In abstract and other text, the abbreviates show the first time should present the whole term, such as BMI (line 13) and CBS (line 18), … et. al.

Yes, absolutely. We have done these changes in the manuscript.

Reviewer 2 Report

The interesting manuscript based on a cross-sectional online questionnaire survey for adolescents [11 to 15 years], attending an international school in Luxembourg (329), and 281 from Paris. However, the details of the questionnaire did not be shown. Finally, the authors concluded “A social control norm was revealed involving a displacement of values affecting body weight and health in the late stage of adolescence to early adolescence, especially for boys.” in the abstract.

Comments:

The title contents “contrasting cultural” from “a French and Luxembourgish context”, however, I did not think the difference of cultural between French and Luxembourgish are the main point in this manuscript. The socioeconomic difference between these 2 groups of adolescents were indeed. Thus, the title and discussion need to re-consider or revise.

The authors deal with the missing data with the data augmentation method in a Bayesian framework. I think need a sensitivity analysis: comparison with and without the missing data augmentation method. I suggest to put the results in the supplementary files.

Minors:

In abstract and other text, the abbreviates show the first time should present the whole term, such as BMI (line 13) and CBS (line 18), … et. al.

Thanks for the opportunity to review the interesting manuscript.

Author Response

2th reviewer. Comments and Suggestions for Authors

The interesting manuscript based on a cross-sectional online questionnaire survey for adolescents [11 to 15 years], attending an international school in Luxembourg (329), and 281 from Paris. However, the details of the questionnaire did not be shown.

Thank you for your proposition. For us, the items inside the questions are in the tables. But if the editor agrees to add the French questionnaire, in the supplementary file, we will do it. Alternatively, questionnaires can be sent upon request to the authors. Concerning the validate Children’s Body Image Scale, you find it in a supplementary material (CBIS); we added an APPENDIX 1 with different pictures of girls and boys.

Finally, the authors concluded ‘A social control norm was revealed involving a displacement of values affecting body weight and health in the late stage of adolescence to early adolescence, especially for boys’ in the abstract.

Thank you

Comments:

The title contents “contrasting cultural” from “a French and Luxembourgish context”, however, I did not think the difference of cultural between French and Luxembourgish are the main point in this manuscript. The socioeconomic difference between these 2 groups of adolescents were indeed. Thus, the title and discussion need to re-consider or revise.

We thank you for this thought full observation. The title has been specified in favour of « in a contrasting socioeconomic milieus, from a French and Luxembourgish context ».

We have revised the words used in the discussions.

The authors deal with the missing data with the data augmentation method in a Bayesian framework. I think need a sensitivity analysis: comparison with and without the missing data augmentation method. I suggest to put the results in the supplementary file.

Thank you for your suggestion. We created at the end of the manuscript an APPENDIX 2 supplementary material with the description of the results and the tables.

Minors: In abstract and other text, the abbreviates show the first time should present the whole term, such as BMI (line 13) and CBS (line 18), … et. al.

Yes, absolutely. We have done these changes in the manuscript.
